# Uncovering the electrical synapse proteome in retinal neurons via in vivo proximity labeling

Stephan Tetenborg[1]*, Eyad Shihabeddin[1†],
Elizebeth Olive Akansha Manoj Kumar[1†], Crystal L Sigulinsky[2†], Karin Dedek[3,4],
Ya-Ping Lin[1], Fabio A Echeverry[5], Hannah Hoff[5], Alberto E Pereda[5],
Bryan W Jones[2], Christophe P Ribelayga[1], Klaus Ebnet[6], Ken Matsuura[7],
John O'Brien[1]*

[1]College of Optometry, University of Houston, Houston, United States; [2]Moran Eye Center/Ophthalmology, University of Utah, Salt Lake City, United States; [3]Animal Navigation/ Neurosensorics, Institute for Biology and Environmental Sciences, University of Oldenburg, Oldenburg, Germany; [4]Research Center Neurosensory Science, University of Oldenburg, Oldenburg, Germany; [5]Dominick P. Purpura Department of Neuroscience, Albert Einstein College of Medicine, Bronx, United States; [6]Institute-Associated Research Group: Cell Adhesion and Cell Polarity, Institute of Medical Biochemistry, ZMBE, University of Münster, Münster, Germany; [7]Cell Signal Unit, Okinawa Institute of Science and Technology, Onna-son, Japan

*For correspondence:
stetenbo@Central.UH.EDU (ST);
jobrien3@central.uh.edu (JO'B)

†These authors contributed equally to this work

Competing interest: The authors declare that no competing interests exist.

## eLife Assessment

This study aims to identify the proteins that make up the electrical synapse, which are much less understood than those of the chemical synapse. These findings represent an **important** step toward understanding the molecular function of chemical synapses and will have broad utility for the wider neuroscience field. The experimental evidence is **convincing**.

**Abstract** Electrical synapses containing Connexin 36 (Cx36) represent the main means for direct electrical communication among neurons in the mammalian nervous system. However, little is known about the protein complexes that constitute these synapses. In the present study, we applied different BioID strategies to screen the interactomes of Connexin 36 and its zebrafish orthologue Cx35.1 in retinal neurons. For in vivo proximity labeling in mice, we took advantage of the Cx36-EGFP strain and expressed a GFP-nanobody-TurboID fusion construct selectively in AII amacrine cells. For in vivo BioID in zebrafish, we generated a transgenic line expressing a Cx35.1-TurboID fusion under control of the *Cx35.1* promoter. Both strategies allowed us to capture a plethora of molecules that were associated with electrical synapses and showed a high degree of evolutionary conservation in the proteomes of both species. Besides known interactors of Cx36 such as ZO-1 and ZO-2, we have identified more than 50 new proteins, such as scaffold proteins, adhesion molecules, and regulators of the cytoskeleton. Moreover, we determined the subcellular localization of these proteins in mouse retina and tested potential binding interactions with Cx36. Among these new interactors, we identified signal-induced proliferation associated 1 like 3 (Sipa1l3), a protein that has been implicated in cell junction formation and cell polarity, as a new scaffold of electrical synapses. Interestingly, Sipa1l3 was able to interact with ZO-1, ZO-2, and Cx36, suggesting a pivotal role in electrical synapse function. In summary, our study provides the first detailed view of the electrical synapse proteome in retinal neurons, which is likely to apply to electrical synapses elsewhere.

## Introduction

With the discovery of acetylcholine as the first neurotransmitter at the beginning of the 20th century, it was widely accepted that chemical synaptic transmission is the only means of communication for neurons. Decades later, several authors independently demonstrated evidence for the existence of electrical synapses in invertebrates as well as in vertebrates, and subsequent studies identified gap junctions, which are composed of clusters of intercellular channels, as the morphological substrates of these synapses (*Bennett, 2000*; *Sotelo, 2020*). Today it is considered common knowledge among neuroscientists that both types of synapses coexist in the nervous system and that each of them provides unique features essential for accurate signal processing. Yet, compared to the efforts that have been made to study protein complexes underlying chemical synapse function and regulation, we know fairly little about the proteome that constitutes an electrical synapse (*Martin et al., 2020*).

In contrast to chemical synapses, electrical synapses don't rely on neurotransmitters and specific receptor molecules but instead make use of intercellular channels to convey neural signals bidirectionally in the form of ionic currents (*Süudhof, 2008*). This seemingly simple mode of communication allows for an instantaneous signal transmission that synchronizes the activities of electrically coupled neurons, giving rise to complex brain functions such as network oscillation, hormone secretion, or coordination of motor functions (*Vaughn and Haas, 2022*; *Hormuzdi et al., 2004*).

In the mammalian nervous system, electrical synapses are mainly, but not exclusively formed by Connexin 36 (Cx36), a connexin isoform that is highly specific for neurons (*Söhl and Willecke, 2004*; *Condorelli et al., 2000*; *Bloomfield and Völgyi, 2009*) and often referred to as the 'major neuronal connexin'. One exception to the rule are the insulin-producing beta cells in the pancreas (*Serre-Beinier et al., 2009*). In general, Cx36 gap junction channels are known to exhibit a very low single-channel conductance (*Srinivas et al., 1999*) but a remarkable degree of plasticity that allows neurons to regulate the number of open channels and hence the strength of synaptic transmission in order to adapt to environmental stimuli (*O'Brien, 2014*). A prime example of a neuron that is heavily coupled via Cx36 and in which the signal transduction pathways that regulate electrical coupling have been well described is the AII amacrine cell (*Kothmann et al., 2012*; *Kothmann et al., 2009*; *Feigenspan et al., 2001*). The AII cell is a glycinergic interneuron in the mammalian retina that is electrically coupled to neighboring AII amacrine cells and to ON Cone bipolar cells (*Marc et al., 2014*). The main task of this cell is to collect neural signals that originate in rods and relay them into the cone pathway. This circuit is referred to as the primary rod pathway, and it forms the neurophysiological basis for scotopic vision. Within the primary rod pathway, electrical coupling of AII amacrine cells is regulated in an activity-dependent manner (*Kothmann et al., 2012*). The opening of Cx36 channels in AII amacrine cells requires the activation of calcium calmodulin-dependent kinase II (CaMKII), which is triggered by the influx of calcium through non-synaptic NMDA receptors. Once CaMKII is activated, it binds to Cx36 and phosphorylates the intracellular domains causing the channel to open (*Kothmann et al., 2012*; *del Corsso et al., 2012*; *Alev et al., 2008*). These activity-driven changes clearly resemble long-term potentiation as it has been described for chemical synapses in hippocampal neurons (*Lisman et al., 2002*). Another, almost obvious, but not less important, parallel between electrical and chemical synapses, is their basic 'architecture': (1) Glutamate receptors, for instance, interact with PSD95 (*Coley and Gao, 2018*), a membrane-associated guanylate kinase (MAGUK) that is similar in function and domain structure to ZO-1, another MAGUK and well-known interactor of Cx36. (2) Electrical synapses are surrounded by adhesions (*Cárdenas-García et al., 2024*). (3) As described for glutamatergic synapses, electrical synapses also harbor electron-dense structures that appear to consist of a dense assemblage of synaptic proteins (*Miller and Pereda, 2017*; *Strettoi et al., 1992*). However, the composition of this density or the type of adhesion molecule that is necessary to specify where an electrical synapse is formed still remains a mystery and requires sophisticated proteomics to solve.

The task of identifying compartment-specific proteomes or interactomes for a given protein of interest (POI) with standard techniques has always been challenging due to the abundance of non-specific background and the requirement to preserve intact protein complexes. To overcome these limitations, several laboratories have developed novel proximity techniques such as TurboID (*Roux et al., 2012*; *Branon et al., 2018*), which utilize promiscuous biotin ligases to biotinylate neighboring proteins within a radius of 5 nm of the POI. Like any other tag, TurboID can be fused to the POI and is then expressed in cells, where it is allowed to biotinylate proximal proteins that are subsequently isolated with streptavidin beads under stringent buffer conditions. The ability to directly

label-'associated' proteins with an affinity tag combined with the fast enzyme kinetics has made TurboID a valuable tool for modern in vivo proteomics. Previously, BioID and TurboID have been used to study the proteomes of chemical synapses and even contact sites of astrocytes and neurons in nematodes and mice (*Takano et al., 2020*; *Artan et al., 2021*; *Uezu et al., 2016*).

In the present study, we have applied two different TurboID approaches to uncover the electrical synapse proteome in the mouse and the zebrafish retina. Contrasting these two model organisms allowed us to expose essential components of the molecular architecture of vertebrate electrical synapses. Beyond known components of electrical synapses, such as ZO-1 and ZO-2, our screen identified a plethora of new molecules that were not previously shown to associate with Cx36/Cx35.1. These include components of the endocytic machinery, adhesion molecules, regulators of the cytoskeleton, membrane trafficking, as well as chemical synapse proteins. Overall, our screen showed that many components of electrical synapses in zebrafish and mouse retinas were highly conserved, in particular the ZO proteins and the endocytosis components, indicating a high degree of evolutionary conservation that suggests a critical functional role for these proteins.

## Results

### Two TurboID approaches allow a glimpse at the electrical synapse proteome in zebrafish photoreceptors and AII amacrine cells of the mouse retina

In this study, we have adopted two in vivo BioID approaches to gain a more comprehensive understanding of protein complexes that are associated with the neuronal connexin Cx36 and its zebrafish orthologue Cx35.1 in retinal neurons. To implement BioID in zebrafish, we generated a transgenic line expressing a Cx35.1-V5-TurboID fusion protein (we used an internal insertion site as previously described *Wang et al., 2015*) under the control of the *Cx35.1* promoter (*Figure 1A*). As shown in control experiments, the expression of this construct recapitulated the expression pattern of wild-type Cx35.1 in photoreceptors (*Li et al., 2009*), which were labeled with ZPR1, indicating that Cx35.1-V5-TurboID is correctly targeted to synapses (*Figure 1B*). The inner retina, however, displayed much weaker expression of the fusion construct, which is why we anticipated that most of the candidates we detected with this strategy are proteins that are associated with Cx35.1 in photoreceptors. To induce efficient biotinylation in zebrafish, we injected 10 mM biotin solution intraperitoneally on three consecutive days and isolated biotinylated proteins for mass spectrometry (*Figure 1B and C*). A complete list of proteins captured in two experiments is shown in *Supplementary file 1*. For the analysis of mass spec data sets, we compared the abundance of candidate proteins from the Cx35.1-V5-TurboID strain to wild-type samples that did not express TurboID, and evaluated specificity using Significance Analysis of Interactome (SAINT) (*Choi et al., 2012*). In Cx35.1-V5-TurboID samples, but not in wild-type retinas, we found several known interactors of the *Gjd2* family such as scaffold proteins, including Tjp1a, Tjp1b, Tjp2a, Tjp2b, Mupp1, Psd95, and Af6 (*Figure 1D*; *Li et al., 2012*; *Tetenborg et al., 2020*; *Lasseigne et al., 2021*; *Ciolofan et al., 2006*). In addition to these candidates, we identified proteins that are involved in membrane trafficking, including Syt4, Sec22b, components of the endosomal trafficking machinery Snap91 and Eps15l1a, and an additional Cx36 orthologue termed Cx34.7, which has previously been localized to photoreceptors in bass retina (*O'Brien et al., 2004*). The detection of this *Gjd1* isoform via BioID was somewhat surprising, as Cx34.7 was shown to form gap junctions that are separated from Cx35.1-containing clusters in bass photoreceptors. To determine the subcellular localization of Cx34.7 and its connection to Cx35.1 in zebrafish photoreceptors, we generated a transgenic Cx34.7-GFP strain that expressed an internally GFP-tagged fusion protein under the control of the *Cx34.7* promoter. In this strain, we found that Cx34.7 was only expressed in the outer plexiform layer (OPL), where it was associated with Cx35.1 at some gap junctions (*Figure 1E*). Thus, besides known interactors of the *Gjd* family, our in vivo BioID approach identified an additional connexin isoform that is connected to Cx35b in photoreceptors.

To unravel the Cx36 interactome in the mouse retina, we initially delivered a Cx36-V5-TurboID fusion via intravitreal AAV injections into retinal neurons. This construct was similarly designed to the Cx35.1 clone but regulated by a short human *SYN*-promoter to drive expression in the neural retina (*Figure 2A*). Unfortunately, this strategy produced massive overexpression artifacts and resulted in an accumulation of Cx36-V5-TurboID in ganglion cell somas, which would have made an interactome

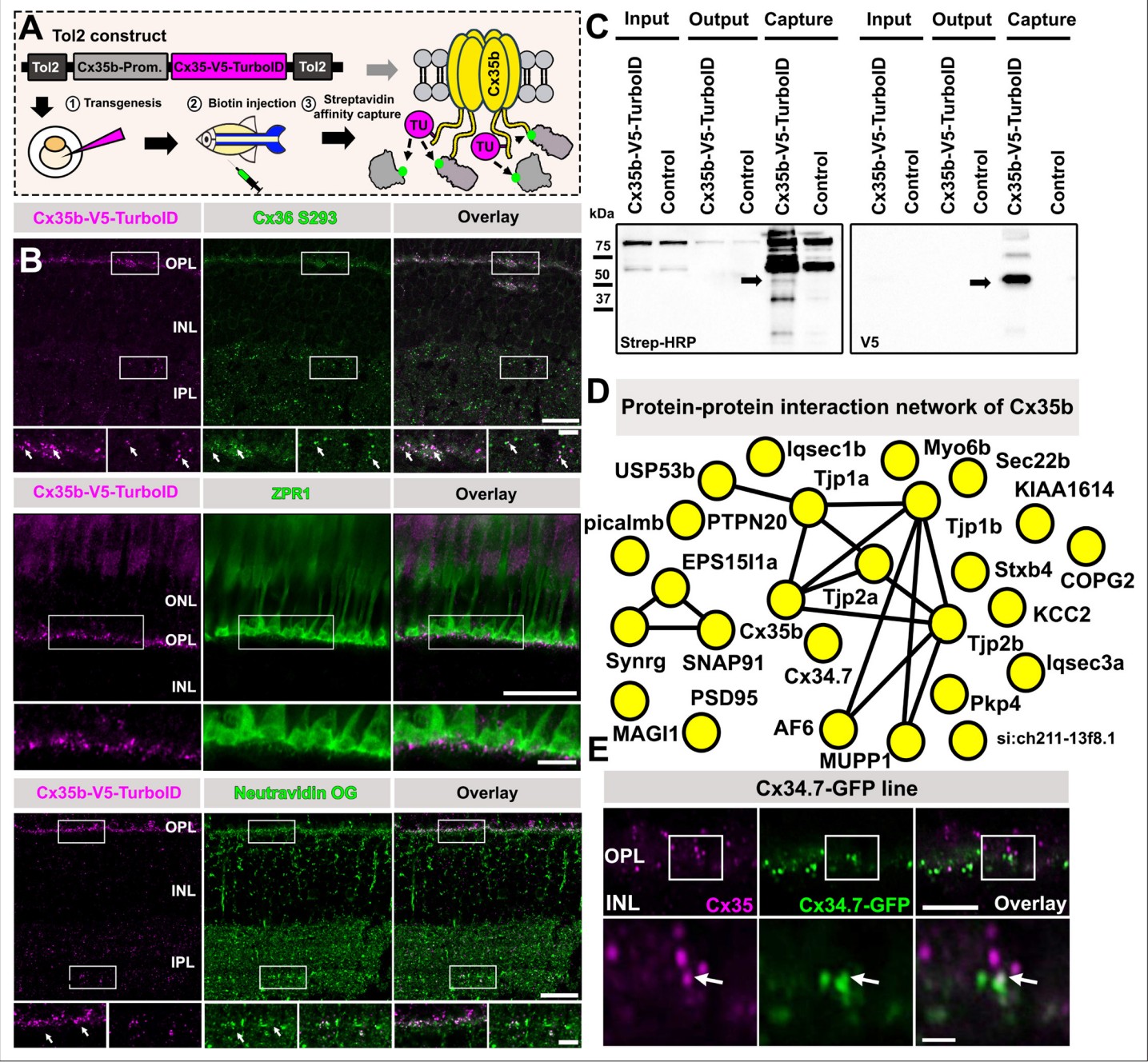

**Figure 1.** Generation of a Cx35.1-V5-TurboID zebrafish line. (**A**) Cartoon illustrating the generation of Cx35.1-V5-TurboID fish via Tol2-mediated transgenesis and outline of in vivo biotinylation experiments. To induce efficient proximity labeling, zebrafish were intraperitoneally injected with 30 μl of 5 mM biotin (PBS) for three consecutive days. Afterward, the animals were sacrificed, and the retinas were isolated for streptavidin pull-downs. (**B**) Confocal scans of entire zebrafish retinas, the outer retina including ZPR1-labeled photoreceptors to confirm successful targeting of Cx35.1-V5-TurboID to photoreceptor gap junctions. Neutravidin Oregon Green labeling was used to validate efficient proximity biotinylation of Cx35.1 and surrounding molecules in biotin-injected Cx35.1-V5-TurboID fish. Reagent or antibody used for labeling is shown in the gray box above each image. Scale 20 μm, magnified inset: 5 μm. (**C**) Western blot of streptavidin pull-downs probed with V5 antibodies and streptavidin-HRP. The arrows indicate the position of the Cx35.1-V5-TurboID construct, which is detected with streptavidin-HRP and a V5 antibody. (**D**) String diagram illustrating the protein-protein network surrounding Cx35. Proteins that were two times or more abundant in comparison to the control condition were included in the string diagram. (**E**) Colocalization of Cx35.1 and Cx34.7-GFP in the outer plexiform layer of the zebrafish retina suggests that a subset of photoreceptor gap junctions contain both connexins. Scale: 5 μm; inset scale: 0.5 μm.

The online version of this article includes the following source data for figure 1:

**Source data 1.** Raw images of western blots shown in *Figure 1*.

**Source data 2.** Raw images of western blots shown in *Figure 1*.

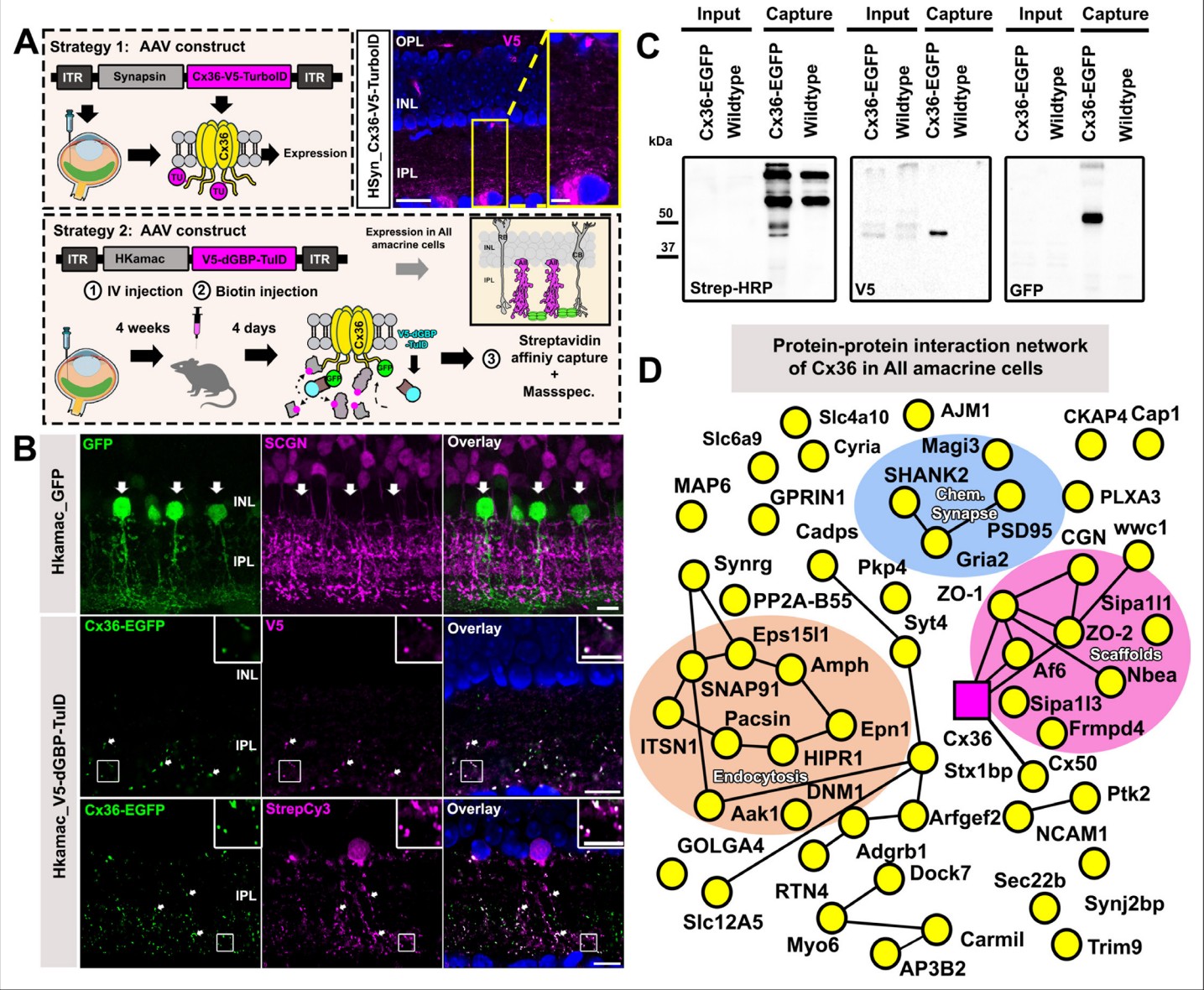

**Figure 2.** A GFP-directed TurboID approach uncovers the Cx36 interactome in AII amacrine cells. (**A**) Cartoon illustrating different in vivo BioID approaches used to target Cx36. In strategy 1, we expressed an AAV Cx36-V5-TurboID construct with an internal insertion site under the control of the human *SYN*-promoter. This strategy greatly resulted in overexpression artifacts causing aberrant protein localization Scale: 20 µm, magnified inset: 5 µm. In strategy 2, we adapted the GFP-directed TurboID strategy developed by *Xiong et al., 2021* to shuttle a V5-dGBP-TurboID construct to Cx36-EGFP containing gap junctions in retinas of the Cx36-EGFP transgenic strain. (**B**) Vertical sections of AAV infected retinas confirming cell type-specificity of our AAV vectors. The HKamac promoter developed by *Khabou et al., 2023* is mainly active in AII amacrine cells and does not show any expression in bipolar cells (SCGN, magenta). V5-dGBP-TurboID colocalizes with Cx36-EGFP in AAV transduced retinas, confirming the successful delivery of TurboID to the gap junction. Scale: 10 µm, magnified inset: 5 µm. (**C**) Western blot confirming successful biotinylation and capture of Cx36-EGFP, biotinylated proteins, and V5-TurboID-dGBP. (**D**) String diagram illustrating the protein-protein interaction network associated with Cx36 in AII amacrine cells. Proteins that were three times or more abundant compared to the control condition were included in the string diagram.

The online version of this article includes the following source data for figure 2:

**Source data 1.** Raw images of western blots shown in *Figure 2*.

**Source data 2.** Raw images of western blots shown in *Figure 2*.

analysis of a synaptic protein impossible. To solve this issue and to ensure that TurboID is targeting electrical synapses, we took advantage of the GFP-directed TurboID approach (*Xiong et al., 2021*) and the Cx36-EGFP transgenic mouse strain (*Helbig et al., 2010*; *Meyer et al., 2014*; *Christie et al., 2005*). The GFP-directed TurboID strategy was initially developed as a modular system for zebrafish by Xiong et al., and utilizes a destabilized GFP nanobody that is fused to TurboID (*Xiong et al., 2021*) to direct the biotin ligase towards a given protein of interest carrying a GFP-Tag. To further increase the specificity of our approach and identify the Cx36 interactome within a single type of inhibitory interneuron, the AII amacrine cell, we expressed V5-TurboID-dGBP under control of the HKamac promoter (*Khabou et al., 2023*). As shown previously, in retinas that were infected with AAVs carrying an HKamac GFP vector, we observed GFP expression mainly in AII amacrine cells but not in bipolar cells (*Figure 2B*, lack of GFP expression in SCGN-labeled neurons). In our hands, we also observed GFP labeling in horizontal cells, which was neglectable for our paradigm, since these neurons do not express Cx36 (*Feigenspan et al., 2004*) and because dGBP is degraded unless it is bound to GFP (*Tang et al., 2016*). In an initial experiment, we injected AAVs carrying HKamac_V5-TurboID-dGBP intravitreally into Cx36-EGFP mice and tracked the localization of the construct two weeks post injection. As expected, all V5 labeled puncta colocalized with Cx36-EGFP, indicating that V5-TurboID-dGBP reached electrical synapses in AII amacrine cells. In the next experiment, we injected a 5 mM biotin solution subcutaneously on four consecutive days, 3 1/2 weeks post infection. This treatment was sufficient to induce efficient biotinylation (*Figure 2B*, lower panel and C) and allowed us to capture a plethora of molecules that were associated with Cx36. To distinguish these proteins from background, we included wild type mice injected with AAV HKamac_V5-TurboID-dGBP throughout the entire experiment. A complete list of proteins captured in two experiments is shown in *Supplementary file 2*. We compared the abundance of all proteins we detected and evaluated specificity using SAINT. A mass spec hit that was three times or even more abundant in the Cx36-EGFP condition and with a SAINT score greater than 0.5 was considered a candidate that is likely to be associated with Cx36. We illustrated the relationship of all proteins that fell into this category in a string diagram (*Figure 2D*). Among the most abundant proteins in our screen, we identified the ZO proteins ZO-1 and ZO-2 and signal induced proliferation associated 1 like 3 (Sipa1l3), a PDZ domain containing protein that has been implicated in the regulation of cell adhesion and cell polarity. In addition to these candidates, we also identified proteins that are involved in endocytosis, membrane trafficking, regulation of the cytoskeleton, cell adhesion, and chemical synapses. All of these hits covered different functional categories of proteins that have been related to connexins (*Flores et al., 2012*; *Lynn et al., 2012*; *Flores et al., 2008*; *Tetenborg et al., 2024a*; *Leithe et al., 2012*; *Martin et al., 2023*) and thus appear to represent an authentic interactome for a neuronal gap junction protein.

## Localization of BioID hits at Cx36 containing gap junctions in the AII amacrine cell

In the next set of experiments, we validated the most abundant candidates from our list in GFP expressing AII amacrine cells. We categorized the proteins based on the cell biological function they serve. In line with the mass spec data, we observed the most frequent colocalization for Cx36 with scaffold proteins including ZO-1, ZO-2, cingulin, and Sipa1l3 (*Figure 3A*, arrows). Although ZO-1, ZO-2, and cingulin are known components of electrical synapses (*Ciolofan et al., 2006*; *Lynn et al., 2012*), Sipa1l3 has not been shown before to associate with Cx36, yet it colocalized with the connexin to a similar extent as the ZO proteins, suggesting that it could be essential for synapse formation or stabilization. In addition to scaffolding proteins, we confirmed an association with several adapter proteins of the endocytic machinery, including Eps15l1, Snap91, Hip1r, and Itsn1 (*Figure 3B*, arrows). In comparison to the scaffold proteins, however, the colocalization of Cx36 with each of these endocytic components was clearly less frequent and more heterogeneous, which appears to reflect different stages in the life cycle of Cx36. As most proteins we identified in this initial screen broadly labeled the entire inner plexiform layer (IPL – the inner synaptic layer of the retina), we performed a colocalization analysis to exclude a random overlap with Cx36. We quantified the number of Cx36 plaques that contained Cgn, Sipa1l3, Eps15l1, and Hip1r (*Figure 3C & D*) and compared the extent of colocalization to a control in which we flipped the Cx36 channel horizontally. For each candidate, we found a significant reduction in colocalization when the original images were compared to the flipped control, indicating that the proteins we have identified are actual components of electrical synapses.

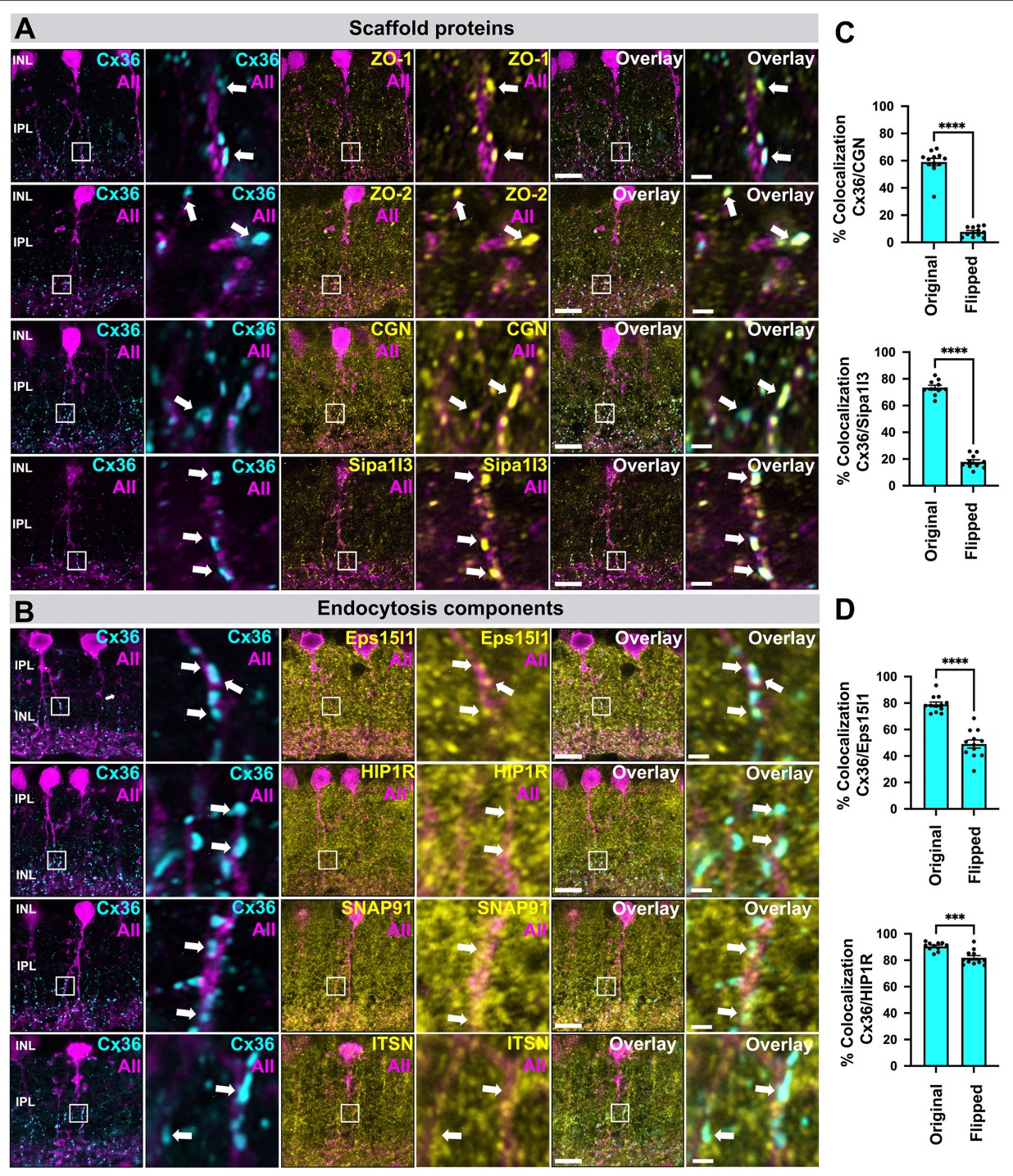

**Figure 3.** Localization of scaffold and endocytosis proteins identified by BioID at Cx36 gap junctions in AAV transduced AII amacrine cells expressing GFP. (**A**) Colocalization of Cx36 and scaffold proteins including known interactors such as ZO-1, ZO-2, and cingulin. Besides these known interactors, we identified Sipa1l3, a PDZ domain containing protein implicated in cell adhesion and cytoskeletal organization. Sipa1l3 shows abundant colocalization with Cx36. (**B**) Colocalization of Cx36 and components of the endocytosis machinery. Among all proteins we have tested, we found frequent colocalization for EpsS15l1, an endocytic adapter protein, and Cx36. Scale: 10 μm. Magnified inset: 1 μm. (**C**) Bar graphs showing the degree of colocalization as the percentage of Cx36 puncta that overlap with Cgn and Sipa1l3. Colocalization was quantified for the original image and a horizontally flipped image to exclude random overlap. Bar graphs with SEM, ***p<0.05 for Cx36 and Cgn and ****p<0.0001 for Cx36 and Sipa1l3. Significance for Cx36 and Cgn was determined using a Wilcoxon matched-pairs signed rank test. Significance for Cx36 and Sipa1l3 was determined using a two-tailed paired t-test. N=10–12 regions of interest. (**D**) Bar graphs showing the degree of colocalization as the percentage of Cx36 puncta that overlap with endocytosis proteins the Ep15l1 and Hip1r. Bar graphs with SEM, ****p<0.0001 for Cx36 and Eps15l1 and ***p<0.05 for Cx36 and Hip1r. Significance was determined using a two-tailed paired t-test. N=10–12 regions of interest.

Besides synaptic scaffolds and endocytic adapter proteins, we tested a variety of additional hits, which include synaptojanin 2 binding protein (Sj2bp), synaptotagmin 4 (Syt4; categorized as 'trafficking') and several proteins that regulate actin dynamics such as G-protein-regulated inducer of neurite outgrowth 1 (Gprin1), dedicator of cytokinesis 7 (Dock7), and microtubule-associated protein 6 (Map6). All of these proteins showed considerable colocalization with Cx36 in AII amacrine cell dendrites (*Figure 4A and B*). We also tested components of chemical synapses including GluR2-3, the scaffold Shank2 (*Figure 4D*), and Bai1 (*Figure 4C*), a synaptic adhesion molecule implicated in excitatory synapse formation (*Tu et al., 2018*). We found that these proteins showed a partial overlap with Cx36 at the periphery of each junction. In addition to the candidates described here, we identified additional candidates that were associated with Cx36 in AII amacrine cells. These include Ajm1 (Apical junction component 1), Cap1 (Adenylyl cyclase-associated protein1), Neurobeachin (Nbea), Ncam1 (Neural cell adhesion molecule 1), Rtn4 (Reticulon4), Af6 (Afadin), Trim9 (Tripartite motif containing protein 9), Golga4 (Golgin subfamily A member 4), Stxbp1 (Syntaxin binding protein 1), and Sipa1l1 (Signal-induced proliferation-associated 1-like protein 1) (*Figure 4—figure supplements 1–5*).

## Identification of novel Cx36 binding partners

Thus far, we have gained significant insight into the Cx36 interactome and identified several new molecules as components of electrical synapses in AII amacrine cells. To understand how these molecules are organized at the synapse, we tested protein–protein interactions in co-transfected HEK293T cells. Interestingly, two of the novel proteins we detected in AII cells contain PDZ domains and thus may directly bind to Cx36 like ZO-1 and ZO-2 (*Figure 5B*, GFP-Trap IP with both ZO proteins served as a positive control); these proteins are Sipa1l3 and Sj2bp. We found that Myc-Sipa1l3 interacted with all major components of electrical synapses: ZO-1, ZO-2, and Cx36 in co-transfected HEK293T cells. The overlap of Sipa1l3 and the ZO proteins was confined to cell cortices and to actin-like fibers in the cytoplasm (*Figure 5A*). Additionally, we performed IP experiments and demonstrated that Cx36 binds to Sipa1l3 in a PDZ-dependent manner. A truncated version of Cx36 lacking the PDZ binding motif (Cx36/S318Ter) failed to interact with Sipa1l3 in IP experiments. In co-transfected cells, we observed that Sj2bp, which contains a single PDZ domain and a transmembrane domain (*Hartmann et al., 2020*), colocalized with Cx36 but not with the Cx36/S318Ter, indicating that the interaction of these proteins requires the PDZ domain. In IP experiments using FLAG-tagged Sj2bp as a bait, we were unable to coprecipitate Cx36, which was rather surprising as both proteins clearly colocalized in co-transfected HEK293T cells. One possible explanation for the lack of a detectable interaction is that Sj2bp displays a low affinity for the PDZ binding motif (PBM) of Cx36, as is the case for ZO-1 (*Flores et al., 2008*). We have previously shown that the binding of Cx36 to ZO-1 can be artificially enhanced by a simple deletion of amino acids 313–319 within the C-terminus, creating a new PBM with the following sequence: RTYV (*Tetenborg et al., 2024b*). We repeated the IP with this mutant and observed substantial binding to FLAG-Sj2bp (*Figure 5B*), suggesting that Cx36 and SJ2BP are compatible interactors. In addition to the candidates we overexpressed to verify direct binding interactions, we found that Cx36 colocalized with several endogenous HEK293T proteins that also occurred in our AII cell-specific data set, including Eps15l1, Gprin1, and Sec22b (*Figure 5C*).

## Molecular architecture of the AII amacrine cell/On cone bipolar cell gap junction

AII amacrine cells form two different sets of gap junctions, with neighboring AII amacrine cells and with ON cone bipolar cells to convey neural signals that originate in rod photoreceptors into the cone pathway (Illustrated in *Figure 6A*; *Bloomfield and Völgyi, 2009*). Ultrastructural studies have revealed an asymmetry in the cytoplasm of AII/ON cone bipolar cell junctions (AII/ONCBC), which exhibits an electron-dense structure (termed fluffy material) in the AII amacrine cell but not in the cone bipolar cell, hinting at differences in the molecular composition of both sites of the synapse (*Strettoi et al., 1992*; *Tsukamoto and Omi, 2017*). Moreover, apart from accessory proteins that bind to the channel, AII/ONCBC gap junctions are likely to be distinct from AII/AII gap junctions since they recognize ON cone bipolar cells as the correct synaptic partner. These two characteristics of the AII/ONCBC gap junction can only be explained by a unique composition of the synapse proteome. We reasoned that the electron density in the AII cell could harbor several proteins we identified in this study. To test this hypothesis, we triple-labeled AAV/HKamac_GFP-infected retinas with Scgn, Cx36, and the main

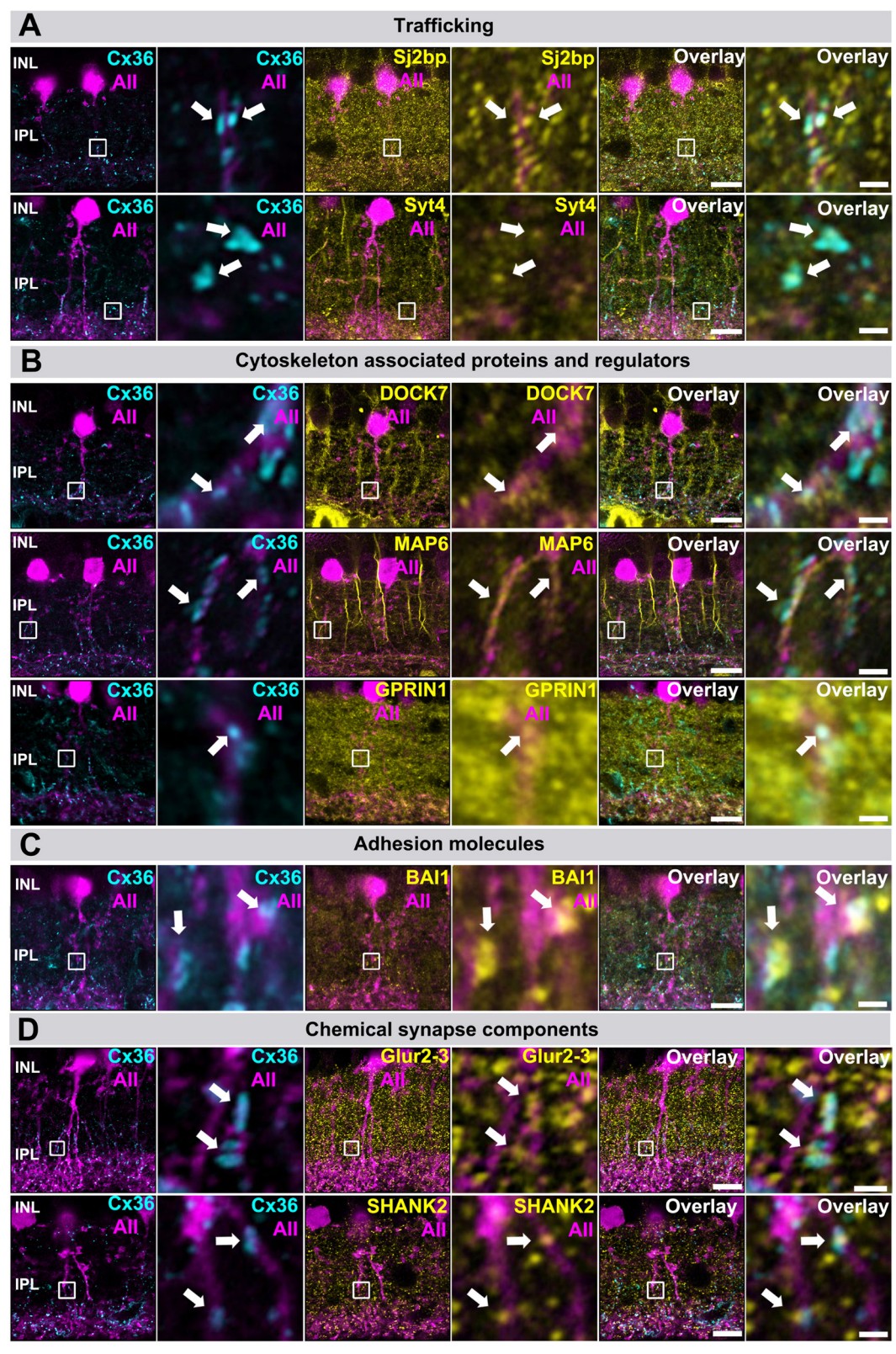

**Figure 4.** Localization of trafficking, cytoskeleton-associated, adhesion, and synaptic proteins identified by BioID at Cx36 gap junctions in AAV transduced AII amacrine cells expressing GFP. (**A**) Proteins implicated in membrane trafficking Sj2bp and Syt4 colocalize with Cx36 in AII amacrine cells. (**B**) Several cytoskeleton-associated proteins and regulators such as Map6, Dock7, or Gprin1 colocalize with Cx36 in AII cell dendrites. (**C**) The adhesion

*Figure 4 continued on next page*

*Figure 4 continued*

molecule Bai1 was often colocalized with Cx36 in AII cell dendrites. (**D**) Often components of chemical synapses such as Shank2 and Glur2-3 were found in the periphery of gap junction plaques in AII amacrine cells. Scale: 10 μm. Magnified inset: 1 μm.

The online version of this article includes the following figure supplement(s) for figure 4:

**Figure supplement 1.** Colocalization of Cx36 and Sipa1l1 in the IPL.

**Figure supplement 2.** Localization of additional proteins captured with BioID.

**Figure supplement 3.** Localization of Ncam1, AF6, Rtn4, and Trim9 cluster in proximity to Cx36 in the inner retina.

**Figure supplement 4.** Localization of Golga4 in the somas of AII amacrine cells.

**Figure supplement 5.** Partial colocalization of Cx36 and Stxbp1 in the IPL.

candidates from our screen. With this strategy, it was possible for us to visualize AII/ONCBC contacts (*Figure 6D*, arrows) and the precise localization of each candidate. To illustrate the localization of AII/ONCBC gap junctions in a comprehensive way, we reconstructed the morphology of AII amacrine cells using serial section transmission EM imaging. As seen in *Figure 6B*, most AII/ONCBC gap junctions (shown in yellow) were located above homologous AII/AII contacts (cyan) and contained electron densities on both sides (*Figure 6C*).

The scaffold proteins ZO-1, ZO-2, Cgn, and Sipa1l3 localized at AII/ONCBC contacts, suggesting that they are a part of the electron density in AII cells (*Figure 6D*, arrows). In addition to these proteins, we also confirmed that AII/Cone bipolar cell gap junctions contained Syt4, Eps15l1, Sj2bp, and Gprin1. Finally, we addressed the exact localization of the G protein-coupled adhesion receptor Bai1, a synaptic adhesion molecule that has been implicated in excitatory synapse formation (*Tu et al., 2018*). Interestingly, we observed Bai1 at AII/ONCBC contacts, in the vicinity of Cx36, which is consistent with the notion of synaptic adhesion molecules as an integral part of electrical synapses (*Cárdenas-García et al., 2024*; *Nagy and Lynn, 2018*). This finding further implies that Bai1 might play a role in the formation of AII/ONCBC contacts. However, further knock-out studies will be necessary to address the concrete function of this adhesion molecule. Finally, we addressed the subcellular distribution of Sipa1l3 at AII/ONBC gap junctions. We noticed that these junctions often displayed a ring-like shape, that contained Sipa1l3 in the center of each ring (*Figure 6E*). One possible interpretation of this distinct sub-synaptic distribution is that Sipa1l3 is interacting with a type of adhesion molecule that is directly adjacent to the gap junction.

## Electrical synapse scaffolds are targeted to AII amacrine cell/ON cone bipolar cell contacts in the absence of Cx36

Previous studies have shown that ZO-1 localizes to neuronal contact sites even in the absence of connexins (*Lasseigne et al., 2021*). This finding is consistent with a study by *Meyer et al., 2014*, who reported that Cx36-EGFP containing gap junctions colocalize with ZO-1 at AII/ONCBC contacts despite a C-terminal tag in the fusion protein that is known to interfere with PDZ domain-mediated interactions (*Meyer et al., 2014*; *Tetenborg et al., 2024b*). One possible interpretation of these observations is that ZO-1, a protein with multiple protein-protein interaction domains, is connected to additional components of the gap junction such as adhesion molecules, the actin skeleton, or other MAGUKs that retain the scaffold at the synapse. We wondered if a similar principle applies to the scaffold proteins that were identified in this study and tested the localization of ZO-2 and Sipa1l3 in Cx36 KO retinas that were infected with AAV/HKamac_GFP to visualize AII/ONCBC contacts. Here, we observed that ZO-2 and Sipa1l3 showed a similar effect as ZO-1 and still localized to AII/ONCBC contacts in Cx36 KO retinas (*Figure 7A–C*). Additionally, we quantified the density and size of immunoreactive Sipa1l3 and ZO-1 in the inner plexiform layer of WT and Cx36 KO mouse retina (*Figure 7D*). While the size of ZO-2 and Sipa1l3 puncta were unchanged in Cx36 KO mice, the density of both ZO-2 and Sipa1l3 puncta were reduced and the colocalization of ZO-2 and Sipa1l3 was reduced in Cx36KO retina. Thus, loss of Cx36 resulted in a quantitative defect in the formation of electrical synapse density complexes.

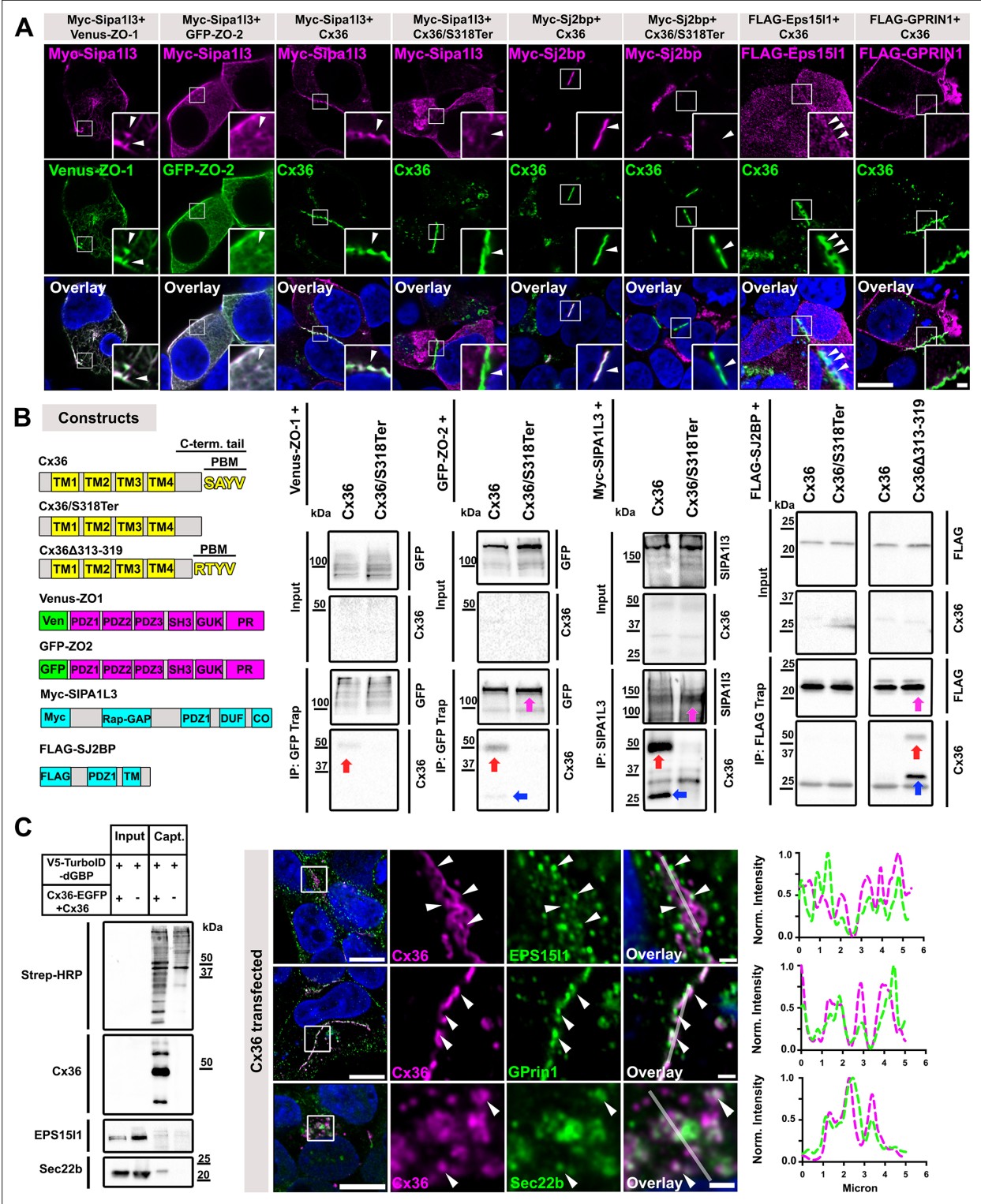

**Figure 5.** Sipa1l3, Sj2bp, and Eps15l1 interact with Cx36. (**A**) Coexpression of Sipa1l3 with ZO-1, ZO-2, and Cx36. In transfected HEK293T cells, Sipa1l3 colocalizes with all three proteins. Colocalization of Sj2bp with Cx36 but not with a Cx36 mutant that lacks the PDZ binding motif. (**B**) Domain organization of constructs used for IP experiments. Co-precipitation of Cx36 and Sjbp2 and Sipa1l3 from lysates of co-transfected HEK293T cells. The PDZ binding deficient Cx36/S318Ter fails to bind Sjbp2 and Sipa1l3. IP experiments with ZO-1 and ZO-2 serve as a positive control. The proteins that were captured in each IP experiment are indicated with arrows in magenta. Cx36 dimers in IP samples are indicated with red arrows and monomers are indicated with blue arrows. (**C**) In transfected HEK293T cells, Cx36 is associated with endogenous proteins that were also detected in AII amacrine cells.

*Figure 5 continued on next page*

*Figure 5 continued*

Line scans next to each panel illustrate the close association of these proteins (**Greenlees et al., 2015**) with Cx36 (magenta). These proteins were also captured via GFP-directed proximity biotinylation. Scale: 10 µm. Magnified inset: 1 µm.

The online version of this article includes the following source data for figure 5:

**Source data 1.** Raw images of western blots shown in *Figure 5*.

**Source data 2.** Raw images of western blots shown in *Figure 5*.

## Discussion

In the present study, we have used TurboID to uncover the electrical synapse proteome in retinal neurons. Our screen identified a plethora of molecules that were associated with the neuronal connexins Cx36/Cx35.1 in zebrafish and mice such as adhesion molecules, scaffold proteins, chemical synapse proteins, components of the endocytic machinery, and cytoskeleton-associated proteins (illustrated in *Figure 7E–F*). The presence of ZO proteins and endocytosis components in the Cx36 and the Cx35.1 interactome data suggests a certain degree of conservation in the proteome of electrical synapses (*Supplementary file 3*). An apparent difference between species, however, is the existence of additional ZO variants and Cx36 homologues in the zebrafish genome (*Lasseigne et al., 2021*), which are also represented in our Cx35b interactome. In contrast to the bass retina, where it was shown that Cx35.1 and Cx34.7 form two distinct circuits (*O'Brien et al., 2004*), we found that zebrafish photoreceptors formed gap junctions that contained both connexins. Cell culture experiments have confirmed that Cx35.1 and Cx34.7 are able to form heterotypic channels (*O'Brien et al., 1998*) that exhibit slightly altered voltage sensitivities compared to the homotypic channels. How exactly these gap junction channels in zebrafish photoreceptors are configured and which exact cell types they connect remains to be determined.

Among other proteins that were detected in this study, we identified Sipa1l3 as a novel scaffold protein for electrical synapses. Interestingly, Sipa1l3 was also described as a component of postsynaptic densities of glutamatergic synapses in hippocampal neurons, and is known to interact with Fezzins, which occur in a complex with Shank3 (*Dolnik et al., 2016*). Additionally, we found that Sipa1l3 was able to interact with Cx36 and the ZO proteins. This raises the question of what potential function Sipa1l3 might serve given the abundance of colocalization with Cx36 in AII amacrine cells? Genetic screens have shown that mutations in the Sipa1l3 gene result in abnormal eye and lens development and a decrease in the formation of cell adhesions. Like other members of the Sipa1l family, Sipa1l3 also contains a RAP GTPase-activating protein (GAP) domain that is known to regulate the activity of RAP proteins, which in the GTP-bound form impacts processes such as adhesion and actin dynamics (*Greenlees et al., 2015*). One way in which Sipa1l3 might influence the formation of electrical synapses might be as a regulator of Rap1 and its effector proteins. To determine the exact function of Sipa1l3, further studies on KO mice will be required.

One striking aspect of the Cx36 interactome in AII amacrine cells is the abundance of the endocytosis machinery. Considering the short half-life time of Cx36 of 3.1 h (*Wang et al., 2015*) and the fact that the turnover of electrical synapse proteins is a steady-state process (*Flores et al., 2012*), it makes sense that the endocytic machinery is that well represented in our interactome data. Thus, besides phosphorylation, it seems possible that the cellular regulation of turnover mechanisms could function as an additional means to adjust the strength of electrical coupling between AII amacrine cells.

### Is Bai1 necessary to form electrical synapses in the primary rod pathway?

How and when it is determined where electrical synapses are formed is currently unknown. It has been suggested that neuronal adhesion molecules serve as cues that wire the correct neurons together (*Martin et al., 2020*). This task also has to be accomplished in the primary rod pathway, where AII amacrine cells form synapses among each other and with ON Cone bipolar cells. Interestingly, we identified the brain-specific angiogenesis inhibitor 1 (Bai1) between AII amacrine and ON cone bipolar cell gap junctions, which could suggest that Bai1 is necessary to connect these cells. Recent reports have described Bai1 as a synaptogenic enzyme capable of regulating intracellular signaling pathways that determine synapse formation (*Tu et al., 2018*; *Duman et al., 2013*). Thus, besides merely functioning as a cell adhesion molecule, Bai1 might actively support synaptogenesis in AII amacrine cells

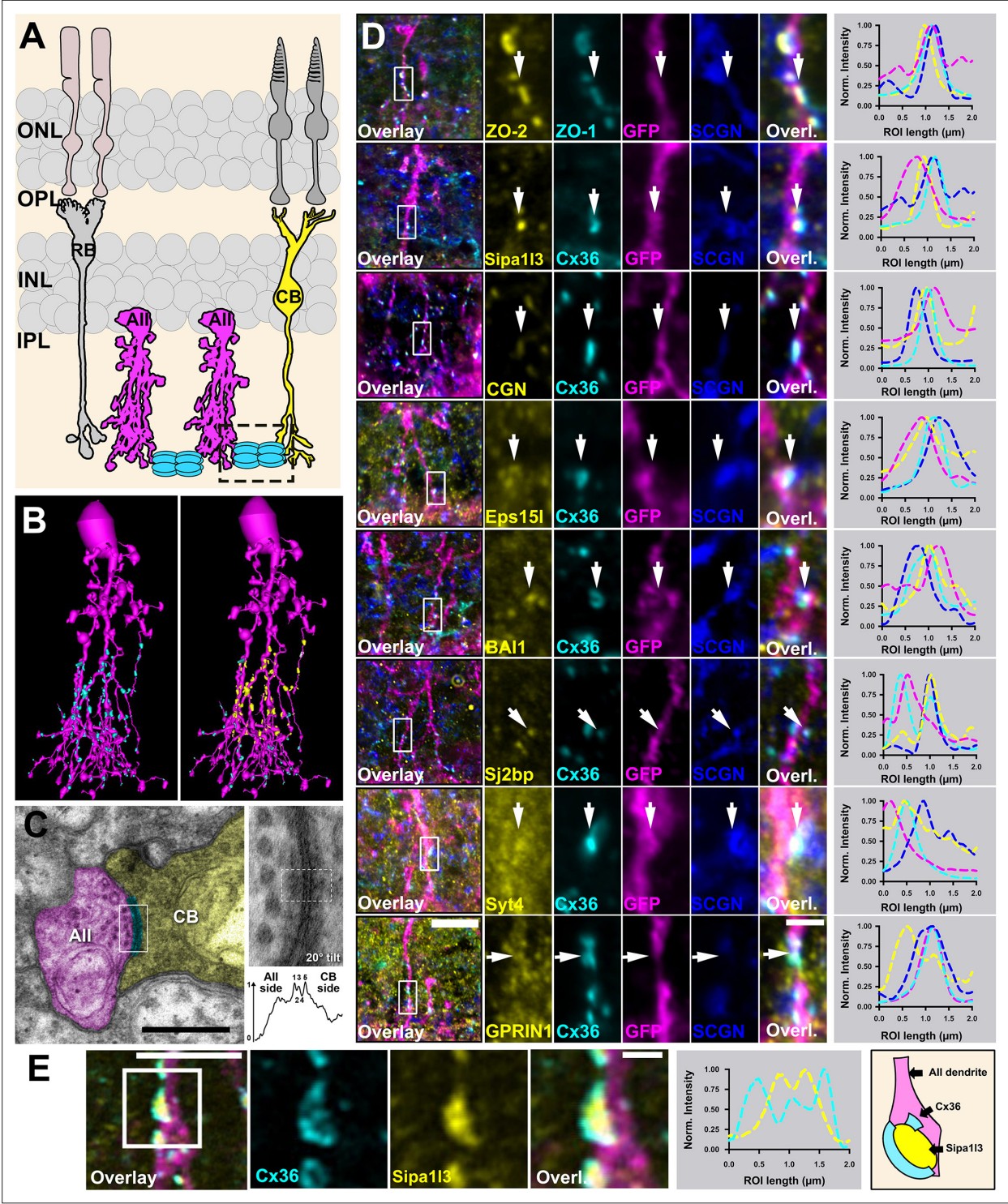

**Figure 6.** Localization of BioID 'hits' at AII amacrine cell/ON cone bipolar cell junctions. (**A**) Cartoon illustrating the neurons involved in the primary rod pathway and the subcellular localization of AII/Cone bipolar cell junctions (dashed rectangle). (**B**) 3D reconstruction of AII amacrine cells from the RC2 connectomics dataset illustrating the localization of AII-AII gap junctions (cyan plaques in second image) and AII/ON Cone bipolar cell gap junctions (yellow plaques in second image). (**C**) EM micrographs of AII/ON Cone bipolar cell gap junctions. Pseudo-colored: Gap junction (cyan), AII cell (magenta), and ON cone bipolar cell (yellow). White box indicates area shown to the right at 40,000× magnification (0.27 nm/px resolution) with 20° tilt. The density profile for the ROI indicated by the yellow box confirms the pentalaminar (dark-light-dark-light-dark) structure of a gap junction. Also note the asymmetric cytoplasmic densities on the AII versus ON Cone bipolar cell sides. Scale: 0.5 µm. (**D**) Several of the proteins we identified in AII amacrine cells colocalize with Cx36 at AII/cone bipolar cell junctions. The right plot for each panel depicts an intensity scan of a horizontal

*Figure 6 continued*

region of interest in the middle of each gap junction (arrow). Scale: 10 µm. Magnified inset: 1 µm. (**E**) High-resolution scan highlighting the subsynaptic distribution of SIPA1L3. Scale: 5 µm. Magnified inset: 1 µm.

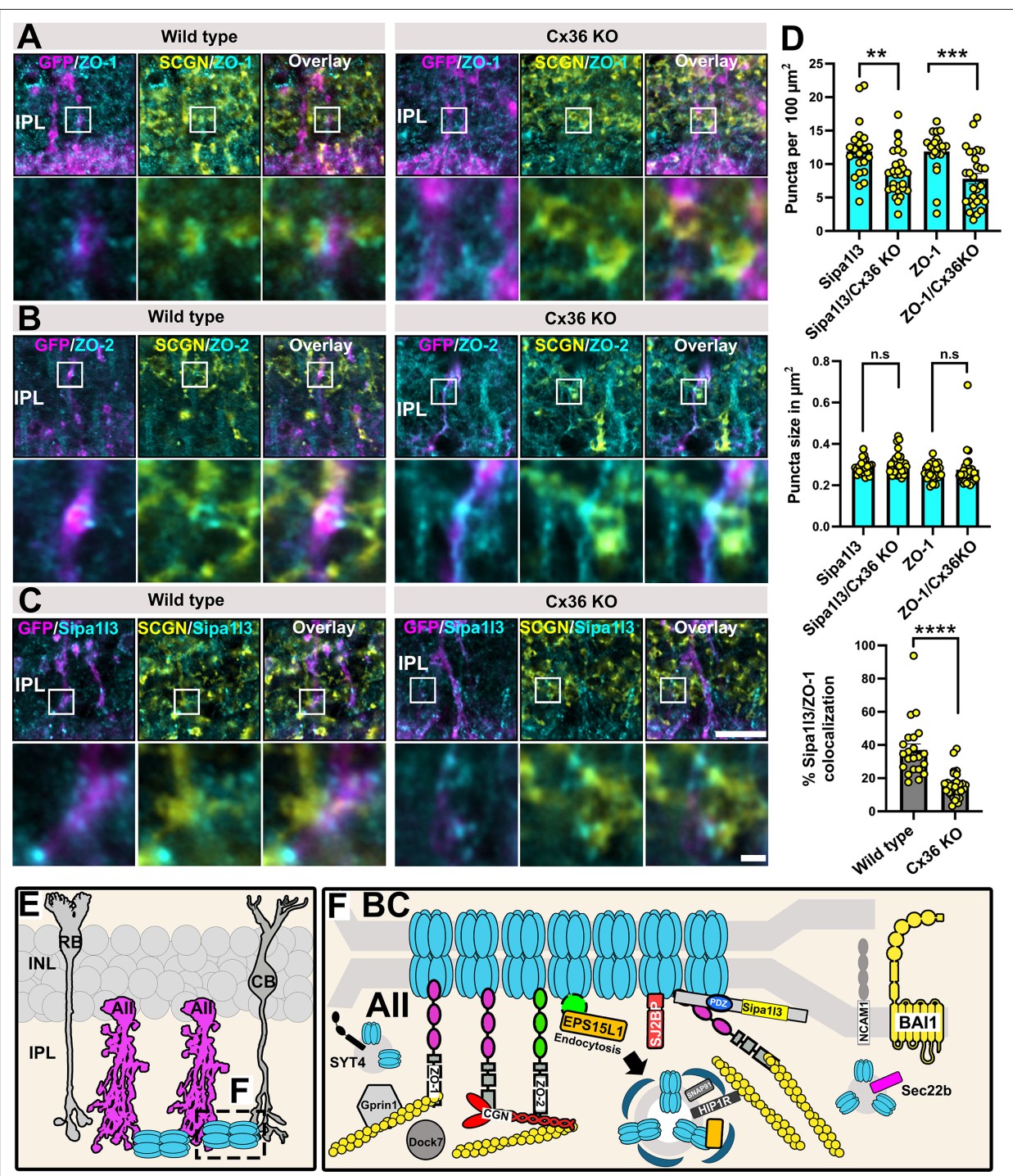

**Figure 7.** AII amacrine cell/ON cone bipolar cell contacts remain in Cx36 KO retinas and contain electrical synapse scaffolds. (**A–C**) AII amacrine cell/ON cone bipolar cell contacts were visualized with GFP and SCGN. In Cx36 KO retinas, ZO-1, ZO-2, and Sipa1l3 still localize to AII/CBC contacts. (**D**) Bar graphs with SEM. Both Sipa1l3 and ZO-1 puncta density in the inner plexiform layer were reduced in Cx36KO, but puncta size was unchanged. Colocalization of Sipa1l3 and ZO-1 was reduced in Cx36KO. For each analysis, 22–32 regions of interest were quantified. (**E, F**) Cartoon illustrating. Scale: 10 µm. Magnified inset: 1 µm.

at a level that is hierarchical to Cx36 and its associated scaffolds, which would explain why these cells form contacts with ON Cone bipolar cells even in the absence of Cx36.

### Cell type-specific differences in the Cx36 interactome?

Our Cx36 interactome in AII amacrine cells showed no signs of the scaffold protein Mupp1, although it was previously identified as a component of electrical synapses in the retina and several other brain regions (*Li et al., 2012*). Interestingly, however, Mupp 1 was detected in our Cx35.1 interactome in zebrafish retina. One feasible explanation is that AII cell gap junctions simply lack certain scaffolds such as Mupp 1 that are used by electrical synapses of other neurons like bipolar cells, which our approach did not consider. In general, it will be interesting to understand how well certain components of the electrical synapse proteome are preserved between cell types. Endocytosis and ZO proteins, for instance, are unlikely to be cell-type-specific interactors of gap junction channels and are more likely to be ubiquitous constituents of electrical synapses, given their critical function in the constant turnover of Cx36. This is also reflected in the evolutionary conservation of these components in the proteomes from zebrafish and mice. Bai1, on the other hand, a protein that is needed to establish specific synaptic contacts, may only occur in certain electrical synapses.

## Materials and methods

**Key resources table**

| Reagent type (species) or resource | Designation | Source or reference | Identifiers | Additional information |
|---|---|---|---|---|
| Antibody | AJM1, rabbit polyclonal | Thermo Fisher | Cat#PA5-145137 RRID:AB_3091817 | 1:100–250 |
| Antibody | BAI1, rabbit polyclonal | Novus Biologicals | Cat# NB110-81586 RRID:AB_1144873 | 1:100 |
| Antibody | Connexin Cx35/36, mouse monoclonal | Millipore | Cat#MAB3045 RRID:AB_94632 | 1:250 |
| Antibody | Connexin 36, mouse monoclonal | Thermo Fisher | Cat#37-4600 RRID:AB_2533320 | 1:250 |
| Antibody | CGN, rabbit polyclonal | Thermo Fisher | Cat# PA5-55661 RRID:AB_2639733 | 1:250 |
| Antibody | Dock7, rabbit polyclonal | Proteintech | Cat#13000-1-AP RRID:AB_10646476 | 1:100 |
| Antibody | EPS15l1, rabbit polyclonal | Thermo Fisher | Cat# PA5-65940 RRID:AB_2665047 | 1:250 |
| Antibody | GFP, chicken polyclonal | Thermo Fisher | Cat# A10262 RRID:AB_2534023 | 1:250 |
| Antibody | GFP, rabbit monoclonal | Cell Signalling | Cat#2956 RRID:AB_10692764 | 1:1000 |
| Antibody | Glur2-3, rabbit polyclonal | Sigma | Cat#07-598 RRID:AB_11213931 | 1:200 |
| Antibody | Gprin1, rabbit polyclonal | Proteintech | Cat#13771-1-AP RRID:AB_2114013 | 1:100 |
| Antibody | HIP1R, rabbit polyclonal | Sigma | Cat# AB9882 RRID:AB_10260185 | 1:250 |
| Antibody | ITSN1, rabbit polyclonal | Thermo Fisher | Cat# PA5-115432 RRID:AB_2900068 | 1:100–250 |
| Antibody | MAP6, rabbit polyclonal | Novus Biologicals | Cat# NBP2-14220 RRID:AB_3261703 | 1:100 |
| Antibody | Myc, rabbit polyclonal | Proteintech | Cat#16286-1-AP RRID:AB_11182162 | 1:1000–2000 |

*Continued on next page*

*Continued*

| Reagent type (species) or resource | Designation | Source or reference | Identifiers | Additional information |
|---|---|---|---|---|
| Antibody | NCAM1, rabbit monoclonal | Cell Signalling | Cat#50831<br>RRID:AB_2868490 | 1:100 |
| Antibody | Neurobeachin, rabbit polyclonal | Thermo Fisher | Cat#PA5-58903<br>RRID:AB_2644492 | 1:100 |
| Antibody | SCGN, sheep polyclonal | Biovendor | Cat# RD184120100<br>RRID:AB_10719237 | 1:500 |
| Antibody | SCGN, rabbit polyclonal | Cusabio | Cat#CSB-PA020821LA01HU | 1:250–500 |
| Antibody | SHANK2, guinea pig polyclonal | Synaptic Systems | Cat#162204<br>RRID:AB_2619861 | 1:500 |
| Antibody | Stxbp1, rabbit polyclonal | Proteintech | Cat#11459-1-AP<br>RRID:AB_2196690 | 1:100 |
| Antibody | Rtn4, rabbit polyclonal | Abcam | Cat#ab47085<br>RRID:AB_881718 | 1:100 |
| Antibody | SIPA1L1, rabbit polyclonal | Proteintech | Cat#25086-1-AP<br>RRID:AB_2714023 | 1:100 |
| Antibody | SIPA1L3, rabbit polyclonal | Proteintech | Cat#0544-1-AP<br>RRID:AB_3086357 | 1:100 |
| Antibody | SJ2BP, rabbit polyclonal | Proteintech | Cat#15666-1-AP<br>RRID:AB_22021149 | 1:100 |
| Antibody | Synaptotagmin4, rabbit polyclonal | Synaptic Systems | Cat#105 143<br>RRID:AB_2619771 | 1:100 |
| Antibody | vGlut1, guinea pig polyclonal | Sigma | Cat#AB5905<br>RRID:AB_2301751 | 1:500 |
| Antibody | V5, mouse monoclonal | Thermo Fisher | Cat#R969-25<br>RRID:AB_2556564 | 1:1000 |
| Antibody | ZO-1, mouse monoclonal | Thermo Fisher | Cat#33-9100<br>RRID:AB_2533147 | 1:250 |
| Antibody | ZO-2, rabbit polyclonal | Thermo Fisher | Cat#71-1400<br>RRID:AB_2533976 | 1:250 |
| Antibody | Anti-rabbit-HRP, goat polyclonal | Thermo Fisher | Cat#34160<br>RRID:AB_228341 | 1:1000 |
| Antibody | Anti-mouse-HRP, goat polyclonal | Thermo Fisher | Cat#34130<br>RRID:AB_228307 | 1:1000 |
| Antibody | Anti-chicken 488, donkey polyclonal | Jackson Immunoresearch | Cat#703-545-155<br>RRID:AB_2340375 | 1:250 |
| Antibody | Anti-rabbit Cy3, donkey polyclonal | Jackson Immunoresearch | Cat# 711-165-152<br>RRID:AB_2340606 | 1:250 |
| Antibody | Anti-mouse Cy5, donkey polyclonal | Jackson Immunoresearch | Cat# 715-175-150<br>RRID:AB_ 2340819 | 1:250 |
| Antibody | Anti-mouse DyLight 405, donkey polyclonal | Jackson Immunoresearch | Cat#715-475-140<br>RRID:AB_2340838 | 1:100–250 |
| Antibody | Anti-guinea pig 647, donkey polyclonal | Jackson Immunoresearch | Cat#706-605-148<br>RRID:AB_2340476 | 1:250 |
| Antibody | Anti-sheep 647, donkey polyclonal | Jackson Immunoresearch | Cat#713-605-003<br>RRID:AB_2340750 | 1:250 |
| Antibody | Anti-mouse 647, donkey polyclonal | Jackson Immunoresearch | Cat#715-605-151<br>RRID:AB_2340863 | 1:250 |

*Continued*

| Reagent type (species) or resource | Designation | Source or reference | Identifiers | Additional information |
|---|---|---|---|---|
| Commercial assay or kit | Dynabeads MyOneStreptavidin C1 | Thermo Fisher | Cat#65001 RRID:SCR_021025 | |
| Commercial assay or kit | ChromoTek GFP-Trap Agarose | Proteintech | Cat#gta RRID:AB_2631357 | |
| Commercial assay or kit | ChromoTek DYKDDDDK Fab-Trap Agarose | Proteintech | Cat#ffa RRID:AB_2631357 | |
| Commercial assay or kit | Pierce Protein A/G Agarose | Thermo Fisher | Cat#20421 | |
| Recombinant DNA reagent | Venus-ZO-1-GFP | Addgene | Cat#56394 | |
| Recombinant DNA reagent | pEGFP-C3-ZO-2-GFP | Addgene | Cat#27422 | |
| Recombinant DNA reagent | FLAG-Eps15l1 | GenScript | Clone: OMu22488D | |
| Recombinant DNA reagent | FLAG-Gprin1 | GenScript | Clone: OMu09339D | |
| Recombinant DNA reagent | AAV_HKamac_V5-dGBP-TurboID | | | Generated in this study |
| Recombinant DNA reagent | AAV_HKamac_GFP | | | Generated in this study |
| Recombinant DNA reagent | AAV_HSyn_Cx36-V5-TurboID | | | Generated in this study |

## Animal husbandry of zebrafish

Maintenance and breeding of zebrafish was conducted under standard conditions (*Westerfield, 2007*). Wild type (AB) zebrafish were purchased from the Zebrafish International Resource Center (ZIRC, Eugene, OR, USA). Fish were maintained on a 14 h light/10 h dark cycle. All procedures were performed in accordance with the ARVO statement on the use of animals in ophthalmic and vision research and US Public Health Service guidelines and have been reviewed and approved by the Institutional Animal Care and Use Committees at the University of Texas Health Science Center at Houston and the University of Houston.

## Tol2-mediated transgenesis

Transgenic Cx35.1-V5-TurboID fish were generated using the Tol2 system and a donor vector containing 4.5 kb of the *Cx35.1* promoter, exon 1 of the *Cx35.1* gene, and the open-reading frame of Cx35.1-V5-TurboID. The coding sequence of V5-TurboID (*Branon et al., 2018*) was cloned into a region of the *Cx35.1* gene encoding the C-terminal tail. V5-TurboID was inserted between leucine 285 and proline 286 to expose the 19 C-terminal amino acids of Cx35.1. Additionally, the donor vector contained a *Myl7* promoter driving mCherry expression in the heart to select larvae in which the construct was integrated into the genome. Transgenic Cx34.7-oxGFP fish were generated in the same way using a donor vector containing 2.8 kb of the *Cx34.7* promoter and the coding sequence of Cx34.7 with oxGFP inserted between aspartate 283 and methionine 284 in the C-terminus. To generate transgenic fish, the donor vector was co-injected with Tol2 transposase mRNA into embryos in the one-cell stage as described previously (*Santhanam et al., 2020*). Transgenic individuals were raised to adulthood and mated.

## Animal husbandry of mice

The experiments in this study were conducted with wildtype mice (C57BL/6J), the Cx36-EGFP strain (*Helbig et al., 2010*; *Meyer et al., 2014*; *Christie et al., 2005*) (kindly provided by Dr. Hannah Monyer) and Cx36 knockout mice (*Jin et al., 2020*) (kindly provided by Dr. David Paul). Animals of ages 2–12 months were used for experiments. All procedures were performed in accordance with the ARVO statement on the use of animals in ophthalmic and vision research and US Public Health Service guidelines and were approved by the Institutional Animal Care and Use Committee at the University of Houston.

## AAV constructs and intravitreal injections

All AAV constructs used in this study were generated with the pAAV-9-(5)-hSYN-CAMKII-GFP vector kindly provided by Dr. Neal Waxman. The coding sequence of the AII amacrine cell specific AII

promoter Hkamac was previously described by *Khabou et al., 2023*, and synthesized as a 654 base pair gBlocks fragment by IDT genomics. The *SYN*-promoter and the coding sequence of CaMKII in the pAAV-9-(5)-*SYN*-CAMKII-GFP vector were excised via XbaI and EcoRV and the coding sequence of V5-TurboID-dGBP (*Xiong et al., 2021*; *Tetenborg et al., 2024b*; *Tetenborg et al., 2023*) and GFP were fused to the AII promoter and integrated into the vector via Gibson assembly. Titers of AAV particles for different AAV constructs varied ranging from $8 \times 10^{12}$ to $1 \times 10^{13}$ gc/ml. For each experimental condition, similar amounts of virus particles were injected. All virus particles were generated at the Baylor College of Medicine gene vector core facility. All AAV vectors that were used in this study will be made available via Addgene. Prior to each virus injection, mice were anesthetized with 3.5–3.9% isoflurane using the SomnoSuite/Mouse STAT Pulse Oximeter (SS-01-03, Kent Scientific Corporation). Once the animal was anesthetized, a small incision next to the pupil was made using a 30 G needle. Afterward, 1.5 µl of the virus solution was carefully injected intravitreally. Eyes were collected 2–4 weeks after each experiment.

## DNA constructs

FLAG-SJ2BP and Myc-SIPA1L3 constructs were used in previous studies (*Hartmann et al., 2020*; *Matsuura et al., 2022*). Venus-ZO-1 (#56394) and GFP-ZO-2 (#27422) were purchased from Addgene. The FLAG-Gprin1 (Clone: OMu09339D) and FLAG-EPS15L1 (Clone: OMu22488D) constructs were purchased from Genscript. All Cx36 pcDNA3.1 expression vectors were generated in previous studies (*Tetenborg et al., 2024b*; *Tetenborg et al., 2023*; *Tetenborg et al., 2024c*).

## In vivo biotinylation

### In vivo biotinylation in zebrafish

Prior to biotin injections, adult zebrafish were anesthetized with 0.02% Tricaine and immobilized in a small wax mold. A small incision was made into the abdomen and 30 µl of a 5 mM biotin (solved in PBS) solution was injected intraperitoneally using a Hamilton syringe. This procedure was repeated for 2 consecutive days. Afterward, the fish were sacrificed by a cold shock and the retinas were extracted. The streptavidin pull down was carried out as previously described (*Tetenborg et al., 2024b*; *Tetenborg et al., 2023*). 80 retinas for each condition (wild type vs. Cx35.1-V5-TurboID) were used. The experiment was performed twice.

### In vivo biotinylation in mouse

3 ½ weeks after intravitreal virus injections, mice were anesthetized with isoflurane and 1 ml of a 5 mM biotin was injected subcutaneously for 4 consecutive days. 17 retinas of each condition (Cx36-EGFP vs. wild type) were used for the streptavidin pull down. The experiment was performed twice.

## LC/MS/MS analysis

For mass spectrometry analysis of streptavidin pull downs, samples were concentrated on an acrylamide/bis-acrylamide stacking gel and processed via In-gel digestion. An aliquot of the tryptic digest (in 2% acetonitrile/0.1% formic acid in water) was analyzed by LC/MS/MS on an Orbitrap Fusion Tribrid mass spectrometer (Thermo Scientific) interfaced with a Dionex UltiMate 3000 Binary RSLC-nano System. Peptides were separated onto an analytical C18 column (100 µm ID × 25 cm, 5 µm, 18 Å Reprosil-Pur C18-AQ beads from Dr. Maisch, Ammerbuch-Entringen, Germany) at a flow rate of 350 nl/min. Gradient conditions were: 3–22% B for 90 min; 22–35% B for 10 min; 35–90% B for 10 min; 90% B held for 10 min (solvent A, 0.1% formic acid in water; solvent B, 0.1% formic acid in acetonitrile). The peptides were analyzed using data-dependent acquisition method; Orbitrap Fusion was operated with measurement of FTMS1 at resolutions 120,000 FWHM, scan range 350–1500 m/z, AGC target 2E5, and maximum injection time of 50 ms; during a maximum 3-s cycle time, the ITMS2 spectra were collected at rapid scan rate mode, with HCD NCE 34%, 1.6 m/z isolation window, AGC target 1E4, maximum injection time of 35 ms. Dynamic exclusion was employed for 20 s.

## Data processing and analysis

The raw data files were processed using Thermo Scientific Proteome Discoverer software version 1.4. Spectra were searched against the Uniprot-Zebrafish or Uniprot-Mouse database using Sequest. Search results were filtered to 1% FDR for strict conditions and 5% for relaxed conditions using

Percolator. Trypsin was set as the enzyme with maximum missed cleavages set to 2. For peptide mapping, MS tolerance was set to 10 ppm, and MS/MS tolerance to 0.6 Da. Carbamidomethylation of cysteine residues was used as a fixed modification, while oxidation of methionine, N-terminal acetylation, and phosphorylation of serine and threonine were set as variable modifications.

## Bioinformatics

For both zebrafish and mouse, data from two experiments with matched experimental and control samples were integrated into a single master table. A mean ratio of Cx35.1/Cx36 conditions to the control was used to identify proteins that were enriched greater than threefold. The data were also analyzed using Significance Analysis of Interactome (SAINT) (*Choi et al., 2012*) implemented via the Resource for Evaluation of Protein Interaction Networks (REPRINT), accessed at https://reprint-apms. org/. Proteins exceeding the threefold enrichment threshold and a SAINT score greater than 0.5 were run through the Cytoscape (V3.10.2) plugin ClueGo (V2.5.10) to identify the GO Biological Processes that were upregulated in each Cx36 condition across both zebrafish and mouse species (*Shannon et al., 2003*; *Bindea et al., 2009*). Pathways that show a p value of less than 0.05 were used for analysis. To determine protein–protein interaction networks, proteins above threshold were also run through the standard STRING (V12.0) pipeline (*Szklarczyk et al., 2023*).

## Cell culture

Human embryonic kidney 293T cells (HEK293T/17; catalog #CRL-11268; ATCC, Manassas, VA, USA) were grown in Dulbecco's Modified Eagle Medium (DMEM) supplemented with 10% fetal bovine serum (FBS), 1% penicillin and streptomycin, and 1% non-essential amino acids (all Thermo Fisher Scientific, Rockford, IL, USA) at 37°C in a humidified atmosphere with 5% $CO_2$. For pull-down experiments, 1 million cells were plated in 60 mm dishes. Transfections were carried out with Geneporter 2 as previously described (*Tetenborg et al., 2024b*).

## Immunocytochemistry

Immunolabeling of transfected HEK293T cells was carried out as previously described (*Tetenborg et al., 2023*; *Tetenborg et al., 2024c*). Transfected HEK293T expressing the proteins of interest were briefly rinsed in PBS and fixed in 2% paraformaldehyde (PFA) solved in PBS for 15 min at room temperature (RT). After the fixation step, the coverslips were washed three times in PBS for 10 min and incubated with the primary antibody solution containing 10% normal donkey serum in 0.5% Triton X-100 in PBS overnight at 4°C. The following antibodies were used: mouse anti-Cx36, 1:500 (MAB3045, Millipore), rabbit anti-SIPA1l3, 1:200 (30544-1-AP, Proteintech), anti-GFP, 1:250 (A10262, Thermo Fisher), rabbit anti-EPS15L1, 1:250 (PA5-65940, Thermo Fisher), mouse anti-V5, 1:500 (R960-2, Thermo Fisher), and rabbit anti-Myc (16286-1-AP, Proteintech). The next day, the coverslips were washed 3 × 10 min in PBS and incubated with the secondary antibodies diluted in 10% normal donkey serum in 0.5% Triton X-100 in PBS at RT under light protected conditions. The following secondary antibodies were used: from donkey, 1:500, conjugated with Cy3, Alexa488, Alexa568, or Alexa647 (Jackson Immunoresearch, West Grove, PA, USA). Afterward, coverslips were washed 3 × 10 min in PBS and mounted with Vectashield containing DAPI (H-2000, Vector Laboratories Inc).

## Immunohistochemistry

Immunolabeling of vertical sections was carried out as previously described (*Tetenborg et al., 2017*; *Tetenborg et al., 2019*). Mice were anesthetized with isoflurane and sacrificed via cervical dislocation. The eyes were removed from the animal and opened with a circular cut around the *ora serrata*. The lens was removed, then eyes were fixed with 2% PFA solved in PBS for 20 min at RT and washed 3 × 10 min in PBS. Afterward, eyecups were stored in 30% sucrose at 4°C overnight. On the next day, eyecup preparations were embedded in Tissue-Tek O.C.T. cryomatrix (Sakura Finetek, Torrance, CA, USA) and stored at –20°C. Eyecups were cut into 20 µm thin sections and incubated at 37°C for 30 min. Afterward, the slides were washed with PBS (2 × 5 min) and incubated in the primary antibody solution containing 10% normal donkey serum in 0.5% Triton X-100 in PBS overnight. The following primary antibodies were used: Cx36, 1:250 (RRID:AB_94632, Clone: 8F6, MAB3045, Millipore); Cx36, 1:250 (RRID:AB_2533320, Clone: 1E5H5, 37-4600, Thermo Fisher); GFP, 1:250 (RRID:AB_2534023, A10262, Thermo Fisher); ZO-1, 1:250 (RRID:AB_3074173, Clone: ZO1-1A12, 33-9100, Thermo

Fisher); ZO-2, 1:250 (RRID:AB_2533976, 71-1400, Thermo Fisher); CGN, 1:250 (RRID:AB_2639733, PA5-55661, Thermo Fisher); SIPA1L3, 1:100 (RRID:AB_3086357, 30544-1-AP, Proteintech); EPS15L1, 1:250 (RRID:AB_2665047, PA5-65940, Thermo Fisher); HIPR1, 1:250 (RRID:AB_10260185, AB9882, Sigma); GluR2-3, 1:200 (RRID:AB_11213931, 07-598, Sigma); Gprin1, 1:100 (RRID:AB_2114013, 13771-1-AP, Proteintech); DOCK7, 1:100 (RRID:AB_10646476, 13000-1-AP, Proteintech); MAP6, 1:100 (RRID:AB_3261703, NBP2-14220, Novus Biologicals); Synaptotagmin4, 1:100 (RRID:AB_2619771, 105143, Synaptic Systems); SJ2BP, 1:100 (RRID:AB_2201149, 15666-1-AP, Proteintech); BAI1, 1:100 (RRID:AB_1144873, NB110-81586, Novus Biologicals); SHANK2, 1:500 (RRID:AB_2619861, Synaptic Systems, 162204); SCGN, 1:250 (CSB-PA020821LA01HU, ARP); SCGN, 1:500 (RRID:AB_10719237, RD184120100, Biovendor); Vglut1, 1:500 (RRID:AB_2301751, AB5905, Sigma); and ITNS1, 1:100-250 (RRID:AB_2900068, PA5-115432, Thermo Fisher). The next day, the slides were washed 3 × 10 min and incubated with secondary antibodies diluted in 10% normal donkey serum in 0.5% Triton X-100 in PBS for 1 h at RT under light protected conditions. The following secondary antibodies were used: from donkey, 1:250, conjugated with Cy3 (RRID:AB_2340813 and RRID:AB_2307443), Alexa488 (RRID:AB_2340375), Alexa568, or Alexa647 (RRID:AB_2340862) or Dylight 405 (RRID:AB_2340839) (Jackson Immunoresearch, West Grove, PA, USA). Afterward, the slides were washed 3 × 10 min and mounted with Vectashield (H-1900, Vector Laboratories, Inc).

## Immunoprecipitation

Immunoprecipitation experiments were carried out as previously described (*Tetenborg et al., 2024a*). 1.5 × 10$^6$ HEK293T cells were plated on 60 mm dishes and co-transfected the next day with 2 µg of each vector using Geneporter2 (Amsbio) transfection reagent. 24 h after transfection, cells were rinsed in phosphate-buffered saline (pH 7.4) and transferred to a reaction tube. The cells were centrifuged at 5000 × *g* for 5 min and afterward homogenized in immunoprecipitation buffer (IP buffer) containing 200 mM NaCl, 2 m EDTA, 1% Tx-100, 50 mM Tris HCl (pH 7.5) supplemented with protease inhibitors (A32965, Thermo Fisher) and 1 mM DTT. The lysate was sonicated 3× for 30 s and incubated on ice for 30 min. Afterward, the lysate was centrifuged at 10,000 × *g* for 10 min. 20 µl of ChromoTek GFP-Trap agarose beads (RRID:AB_2631357, gta, Proteintech) were applied to the supernatant and incubated overnight at 4°C on a rotating platform. The next day the sample was centrifuged at 2500 × *g* for 5 min and washed three times in IP buffer. Bound proteins were eluted with 60 µl of 1× Laemmli buffer (Bio-Rad, 1610747) for 5 min at 95°C. IP experiments for Myc-Sipa1l3 and FLAG-Sj2bp were conducted under a slightly modified protocol. For IP experiments with Myc-Sipa1l3 0.5 µg of Sipa1l3 antibody (30544-1-AP, Proteintech) was applied to the cell lysate and incubated overnight on a rotating platform. The next day, 40 µl of prewashed protein A/G beads (20421, Pierce) were applied to the cell lysate and incubated for 1 h at 4°C. Afterward, the beads were washed three times and bound proteins were eluted with 1× Laemmli buffer. IP experiments with FLAG-Sj2bp were performed with 20 µl ChromoTek DYDDDDK Fab-Trap (RRID:AB_2894836, ffa, Proteintech). Prior to the IP, cell lysates were centrifuged at 20,000 × *g* for 30 min at 4°C. Afterward, cell lysates were incubated with agarose overnight and washed three times at 4°C. Bound proteins were eluted with 60 µl of 1× Laemmli buffer.

## Western blot

Protein samples were separated via SDS-PAGE containing 7.5% (for Sipa1l3 blots) or 10% polyacrylamide. Proteins were transferred to nitrocellulose membranes using the transblot turbo system as previously described (*Tetenborg et al., 2024b*). Blots were blocked in 5% dried milk powder in TBST (20 mM Tris pH 7.5, 150 mM NaCl, 0.2% Tween 20) for 30 min at RT and incubated overnight in the primary antibodies diluted in blocking solution at 4°C on a rotating platform. The following primary antibodies were used: Cx36, 1:500 (Clone: 8F6, MAB3045, Millipore); Cx36, 1:500 (Clone: 1E5H5, 37-4600, Thermo Fisher), SIPA1L3, 1:4000 (30544-1-AP, Proteintech), GFP, 1:1000 (RRID:AB_1196615, 2956, Cell Signaling), and V5 1:1000 (RRID:AB_2556564, R960-25, Thermo Fisher). Biotinylated proteins detected with high-sensitivity streptavidin-HRP, 1:10,000 (21130, Pierce). The next day, blots were washed 3 × 10 min in TBST and incubated with the secondary antibodies, diluted in the blocking solution as described previously (*Tetenborg et al., 2024b*). The following secondary antibodies were used: goat anti-mouse HRP, 1:1000 (RRID:AB_228307, 34130) and goat anti-rabbit, 1:1000 (RRID:AB_228341, 34160, Thermo Fisher). Afterward, the blots were washed 3 × 10 min in TBST and

incubated with the ECL solution (32106, Thermo Fisher) for detection. ECL was detected with the Biorad ChemicDoc MP Imaging System.

## Confocal microscopy and image analysis

Fluorescence images were acquired with a confocal laser scanning microscope (Zeiss LSM 800) using a ×60 oil objective and the Airy scan function (pixel size: 50 nm). Confocal scans were processed using Fiji (ImageJ2 version 2.14.0/1.54f) software (*Schindelin et al., 2012*). The threshold was adjusted using the *Triangle* function for the entire z-stack (seven slices; 1.08 µm). Two regions of interest (ROI) with the size of 20.07 × 32.91 µm$^2$ were positioned to cover the ON and OFF IPL layers. Frequency and size of the immunoreactive puncta were measured for each channel using the 'analyze particles' function. Colocalization of ZO-1 and Sipa1l3 was quantified using the *Colocalization* plugin in Fiji. The number of colocalized puncta and the colocalization was measured using the *Analyze* particle function with a size-exclusion criterion of >100$^2$ pixels. Following the quantification, the size, density, and percentage of the colocalized puncta were averaged for each protein.

## Volume construction

Retinal Connectome 2 (RC2) is an ultrastructural dataset acquired from the retina of a light-adapted 5-month-old female C57BL/6J mouse (*Sigulinsky et al., 2024*). Methods concerning tissue acquisition and processing, volume assembly, visualization, and annotation following that extensively detailed for RC1 (*Anderson et al., 2009*; *Anderson et al., 2011*). In short, the RC2 dataset spans the ganglion cell layer through outer nuclear layer of a 0.25-mm-diameter field of retina. RC2 was constructed from 1335 serial TEM sections (70 nm thick), captured at 2.18 nm/pixel via Automated Transmission Electron Microscopy at 5000× and combined with 31 optical sections intercalated throughout the volume. These optical sections were probed for small-molecule signals for computational molecular phenotyping (*Marc et al., 1995*). Sections were aligned into a single volume using the NCR ToolSet, which has since been replaced by Nornir (RRID:SCR_016458). All protocols were in accord with Institutional Animal Care and Use protocols of the University of Utah, the ARVO Statement for the Use of Animals in Ophthalmic and Visual Research, and the Policies on the Use of Animals and Humans in Neuroscience Research of the Society for Neuroscience.

## Statistical analysis

Data sets acquired in this study were analyzed with GraphPad Prism 8. Data are shown as mean ±SEM. Normality was tested using the Anderson–Darling and the D'Agostino–Pearson test. Significance was tested using the two-tailed Mann–Whitney U test. For multiple comparisons, a one-way ANOVA was performed.

## Materials availability

DNA constructs generated in this study will be made available upon request to Dr. Stephan Tetenborg and Dr. John O`Brien.

## Acknowledgements

This project was supported by NIH grants R01EY012857 (JO), RF1MH120016 (AEP, JO), and R21NS085772 (AEP, JO) and core grant P30EY007551. ST was funded by the Deutsche Forschungsgemeinschaft (DFG) (TE 1459/1-1, Walter Benjamin stipend). ES was supported by NIH training grant TL1TR003169 and individual grant F31EY034793. BWJ and CLS are supported by NIH grants R01EY028927 (to BWJ) and P30EY014800 (to the Moran Eye Center Core); NSF grant 2014862; an Unrestricted Research Grant from Research to Prevent Blindness, New York, NY, to the Department of Ophthalmology & Visual Sciences, University of Pittsburgh; a Stein Innovation Award from Research to Prevent Blindness, New York, NY, and an Unrestricted Research Grant from Gabe Newell (to BWJ). This work is supported in part by the Clinical and Translational Proteomics Service Center at the University of Texas Health Science Center. We would like to thank Li Li from the UT health mass spec. facility. We would like to thank Dr. Adam C Miller and Dr. Jen Michel for the helpful discussions. We appreciate Nikki Brantley's support in the cell culture lab.

# Additional information

## Funding

| Funder | Grant reference number | Author |
|---|---|---|
| Deutsche Forschungsgemeinschaft | TE 1459/1-1 | Stephan Tetenborg |
| NIH Blueprint for Neuroscience Research | R01EY012857 | John O'Brien |
| NIH Blueprint for Neuroscience Research | RF1MH120016 | Alberto E Pereda |
| NIH Blueprint for Neuroscience Research | R01EY028927 | Bryan W Jones |
| National Science Foundation | 2014862 | Bryan W Jones |
| NIH Blueprint for Neuroscience Research | R21NS085772 | Alberto E Pereda John O'Brien |
| NIH Core grant | P30EY007551 | |
| NIH training grant | TL1TR003169 | Eyad Shihabeddin |
| NIH Individual grant | F31EY034793 | Eyad Shihabeddin |
| NIH Blueprint for Neuroscience Research | P30EY014800 | Bryan W Jones |
| Gabe Newell | | Bryan W Jones |

The funders had no role in study design, data collection and interpretation, or the decision to submit the work for publication.

## Author contributions

Stephan Tetenborg, Conceptualization, Data curation, Formal analysis, Supervision, Funding acquisition, Investigation, Methodology, Writing – original draft, Writing – review and editing; Eyad Shihabeddin, Formal analysis, Investigation; Elizebeth Olive Akansha Manoj Kumar, Data curation, Formal analysis, Investigation; Crystal L Sigulinsky, Data curation, Investigation, Methodology; Karin Dedek, Resources, Methodology, Project administration; Ya-Ping Lin, Fabio A Echeverry, Hannah Hoff, Klaus Ebnet, Ken Matsuura, Resources; Alberto E Pereda, Resources, Supervision, Funding acquisition, Writing – review and editing; Bryan W Jones, Resources, Supervision, Funding acquisition; Christophe P Ribelayga, Resources, Supervision; John O'Brien, Conceptualization, Resources, Formal analysis, Supervision, Funding acquisition, Project administration, Writing – review and editing

## Author ORCIDs

Stephan Tetenborg ⓘD https://orcid.org/0000-0002-9025-0446
Eyad Shihabeddin ⓘD https://orcid.org/0000-0002-3140-1523
Elizebeth Olive Akansha Manoj Kumar ⓘD https://orcid.org/0009-0000-1264-3642
Crystal L Sigulinsky ⓘD https://orcid.org/0000-0002-5642-8474
Karin Dedek ⓘD https://orcid.org/0000-0003-1490-0141
Fabio A Echeverry ⓘD https://orcid.org/0000-0002-4200-4080
Alberto E Pereda ⓘD https://orcid.org/0000-0002-8283-8768
Bryan W Jones ⓘD https://orcid.org/0000-0001-5527-6643
Christophe P Ribelayga ⓘD https://orcid.org/0000-0001-5889-2070
Klaus Ebnet ⓘD https://orcid.org/0000-0002-0417-7888
Ken Matsuura ⓘD https://orcid.org/0000-0003-0247-3867
John O'Brien ⓘD https://orcid.org/0000-0002-0270-3442

## Ethics

This study was performed in strict accordance with the recommendations in the Guide for the Care and Use of Laboratory Animals of the National Institutes of Health. All procedures were approved by

the IACUC, Protocol number: PROTO202100038. Accordingly, all surgeries (Intravitreal AAV injections) were performed with isoflurane-induced anesthesia to minimize suffering.

Reviewer #1 (Public review): https://doi.org/10.7554/eLife.105935.4.sa1
Reviewer #3 (Public review): https://doi.org/10.7554/eLife.105935.4.sa2
Author response https://doi.org/10.7554/eLife.105935.4.sa3

# Additional files

## Supplementary files
Supplementary file 1. Excel workbook containing ranked list of all proteins that were detected via mass spectroscopy in Cx35.1-V5-TurboID fish in two experiments, plus SAINT analysis of these experiments.

Supplementary file 2. Excel workbook containing ranked list of all proteins that were detected via mass spectroscopy in Cx36-EGFP-directed proximity labeling in two experiments, plus SAINT analysis of these experiments.

Supplementary file 3. Comparison of electrical synapse components that were identified in zebrafish and mouse retinas.

## Data availability
The mass spectrometry data that were collected in this study are available as supplementary files. Confocal and gel images are stored in a data repository and can be found at https://www.ebi.ac.uk/biostudies/bioimages/studies/S-BIAD3071.

The following dataset was generated:

| Author(s) | Year | Dataset title | Dataset URL | Database and Identifier |
| --- | --- | --- | --- | --- |
| Tetenborg S | 2026 | Uncovering the electrical synapse proteome in retinal neurons via in vivo proximity labeling | https://www.ebi.ac.uk/biostudies/bioimages/studies/S-BIAD3071 | Bioimage Archive, S-BIAD3071 |

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
