## [Editor Report · eLife Assessment]

This study aims to identify the proteins that make up the electrical synapse, which are much less understood than those of the chemical synapse. These findings represent an **important** step toward understanding the molecular function of chemical synapses and will have broad utility for the wider neuroscience field. The experimental evidence is **convincing**.

---

## [Referee Report · Reviewer #1 (Public review)]

This study aims to identify the proteins that compose the electrical synapse, which are much less understood than those of the chemical synapse. Identifying these proteins is important to understand how synaptogenesis and conductance are regulated in these synapses.

Using a proteomics approach, the authors identified more than 50 new proteins and used immunoprecipitation and immunostaining to validate their interaction of localization. One new protein, a scaffolding protein (Sipa1l3), shows particularly strong evidence of being an integral component of the electrical synapse. The function of Sipa1l3 remains to be determined.

Another strength is the use of two different model organisms (zebrafish and mice) to determine which components are conserved across species. This approach also expands the utility of this work to benefit researchers working with both species.

The methodology is robust and there is compelling evidence supporting the findings.

Comments on revisions:

I thank the authors for responding to the comments. No further recommendations.

---

## [Referee Report · Reviewer #3 (Public review)]

Summary:

This study by Tetenborg S et al. identifies proteins that are physically closely associated with gap junctions in retinal neurons of mice and zebrafish using BioID, a technique that labels and isolates proteins in proximal to a protein of interest. These proteins include scaffold proteins, adhesion molecules, chemical synapse proteins, components of the endocytic machinery, and cytoskeleton-associated proteins. Using a combination of genetic tools and meticulously executed immunostaining, the authors further verified the colocalizations of some of the identified proteins with connexin-positive gap junctions. The findings in this study highlight the complexity of gap junctions. Electrical synapses are abundant in the nervous system, yet their regulatory mechanisms are far less understood than those of chemical synapses. This work will provide valuable information for future studies aiming to elucidate the regulatory mechanisms essential for the function of neural circuits.

Strengths:

A key strength of this work is the identification of novel gap junction-associated proteins in AII amacrine cells and photoreceptors using BioID in combination with various genetic tools. The well-studied functions of gap junctions in these neurons will facilitate future research into the functions of the identified proteins in regulating electrical synapses.

Comments on revisions:

The authors have addressed my concerns in the revised manuscript.

---

## [Author Response]

The following is the authors’ response to the previous reviews

**Reviewer 1**
The authors should clarify the statement regarding the expression in horizontal cells (lines 170-172). In line 170, it is stated that GFP was observed in horizontal cells. Since GFP is fused to Cx36, the observation of GFP in horizontal cells would suggest the expression of Cx36-GFP.

We believe that there appears to be a misunderstanding. GFP is observed in horizontal cells, because the test AAV construct, which consists of the HKamac promote**r** and a downstream GFP sequence, was used to validate the promoter specificity in wildtype animals. This was just a test to confirm that HKamac is indeed active in AII amacrine cells as previously described by Khabou et al. 2023. This construct was not used for the large scale BioID screen. For these experiments, V5-dGBP-Turbo was expressed under the control of the HKamac promoter as illustrated in Figure 2A.

Fig 7: the legend is missing the descriptions for panels A-C.

We apologize for this mistake. We have missed the label “(A-C)” and added it to the legend.

Supplemental files are not referenced in the manuscript.

We have added a reference for these files in line 221-226.

**Reviewer 2**
Supplementary Files 1 and 2 are presented as two replicates of the zebrafish proteomic datasets, but they appear to be identical.

This appears to be a misunderstanding. These two replicates contain slightly different hits, although the most abundant candidates are identical.

**Reviewer 3**

Thank you for the positive comments